# Protective Effect of the Phycobiliproteins from *Arthrospira maxima* on Indomethacin-Induced Gastric Ulcer in a Rat Model

**DOI:** 10.3390/plants12081586

**Published:** 2023-04-08

**Authors:** Oscar Guzmán-Gómez, Rosa Virginia García-Rodríguez, Salud Pérez-Gutierrez, Nora Lilia Rivero-Ramírez, Yuliana García-Martínez, Saudy Saret Pablo-Pérez, Ricardo Pérez-Pastén-Borja, José Melesio Cristóbal-Luna, Germán Chamorro-Cevallos

**Affiliations:** 1Departamento de Farmacia, Escuela Nacional de Ciencias Biológicas, Instituto Politécnico Nacional, Av. Wilfrido Massieu 399, Mexico City C.P. 07738, Mexico; oguz1985@live.com.mx (O.G.-G.); ygarciamart@hotmail.com (Y.G.-M.); spablop@ipn.mx (S.S.P.-P.); pastenrich@yahoo.com.mx (R.P.-P.-B.); 2Instituto de Química Aplicada, Universidad Veracruzana, Luis Castelazo Ayala S/N Col. Industrial Ánimas, Xalapa C.P. 91190, Mexico; rosga74@hotmail.com; 3Departamento de Sistemas Biológicos, Universidad Autónoma Metropolitana-Xochimilco, Calzada del Hueso 1100, Mexico City C.P. 04960, Mexico; msperez@correo.xoc.uam.mx; 4Departamento de Morfología, Escuela Nacional de Ciencias Biológicas, Instituto Politécnico Nacional, Prolongación de Carpio y Plan de Ayala s/n, Mexico City C.P. 11340, Mexico; jazzband19@hotmail.com

**Keywords:** spirulina (*Arthrospira maxima*), phycobiliproteins, antiulcerogenic, indomethacin, oxidative stress

## Abstract

Gastric ulcers (GU) constitute a disease with a global prevalence ≈ 8.09 million. Of their causes, non-steroidal anti-inflammatory drugs (NSAIDs) such as indomethacin (IND) rank as the second most frequent etiologic agent. The pathogenic process of gastric lesions is given by the overproduction of oxidative stress, promotion of inflammatory processes, and inhibition of prostaglandin synthesis. Spirulina *Arthrospira maxima* (SP) is a cyanobacterium with a wide variety of substances with high nutritional and health values such as phycobiliproteins (PBPs) that have outstanding antioxidant activity, anti-inflammatories effects, and accelerate the wound healing process. This study aimed to determine the protective effect of PBPs in GU induced by IND 40 mg/kg. Our results show that the PBPs protected against IND-induced damage with a dose-dependent effect. At a dose of 400 mg/kg, a marked decrease in the number of lesions is observed, as well as the recovery of the main markers of oxidative stress damage (MDA) and antioxidant species (SOD, CAT, GPx) at close to baseline levels. The evidence derived from the present investigation suggests that the antioxidant effect of PBPs, together with their reported anti-inflammatory effects to accelerate the wound healing process, is the most reliable cause of their antiulcerogenic activity in this GU model.

## 1. Introduction

A peptic ulcer is a lesion on the gastrointestinal tract (GI) that, even though these ulcerations and erosions may involve the lower esophagus, distal duodenum, or jejunum, usually develops in the stomach and proximal duodenum [1]. When injuries occur inside the stomach, they are named gastric ulcers (GU) while if they appear on the duodenum they are called duodenal ulcers (DU) [2].

A sore that develops inside the stomach can extend into the muscularis propria layer of the gastric epithelium as a result of an imbalance between aggressive factors and mucosal defense barriers, characterized by discontinuation in the inner lining of the gastrointestinal tract because of gastric acid secretion or pepsin [3], known as peptic ulcer disease (PUD) [4]. Although in GU the common symptom is stomach pain, because of the damage to the mucosa and GI epithelium, gastric ulcers can provoke various signs and symptomatology such as epigastric abdominal pain, burning stomach pain, bloating, nausea and vomiting, hematemesis, melena, feeling faint, and weight loss and dyspepsia, between others [5,6,7].

PUD was considered a health problem worldwide from 1990 due to its already high prevalence increasing from 6,434,103 (95% uncertainty interval 5,405,963 to 7,627,971) to 8,090,476 (6,794,576 to 9,584,000) in 2019 [8]. Recent data indicate that PUD affects around four million people annually, with an estimated lifetime prevalence of 5–10% [9] which can evolve into a more severe condition and lead to death. In this sense, at the worldwide population, PUD owns a mortality of 18.2% [10] and a risk of 5 to 10% of developing it throughout life [11].

Although morbidity and mortality by PUD decreased from 1990 to date, the decreasing tendency in recent years has plateaued due to changes in the incidence of risk factors such as *Helicobacter pylori* (*H. pylori*) infection, nonsteroidal anti-inflammatory drugs (NSAIDs) use, selective serotonin reuptake inhibitor use, gastric bypass surgery, genetic characteristics, bad lifestyle habits, smoking, etc. [12].

Over several decades the etiology of GU was studied and it was found that, although it is considered as the result of an imbalance between defensive (cellular regeneration, mucus-bicarbonate layer, mucosal blood flow, and prostaglandin action) and aggressive factors (pepsin, ethanol, stomach acid, bile salts, and drugs), the infection with *H. pylori*, the stress, and the chronic use NSAIDs represent the main causes of the disease [13,14]. After infection with *H. pylori*, the use of NSAIDs is considered the second most common cause of GU [12,15]. Under normal conditions, the rich stomach secretion of mucus stimulated by prostaglandins (PG) provides wide protection to the gastric mucosa against the damage of caustic agents [16]. PG are a series of hormone-like chemical messengers that play a critical role in regulating physiological activity. They are synthesized in the body in three main phases: (I) release of arachidonic acid from membrane phospholipids by phospholipase-A2; (II) oxidation of arachidonic acid by the isoforms 1 and 2 of the cyclooxygenase (COX) to PG G2 (PGG2), then peroxidized to PGH2; and (III) PGH2 is transformed into five main PG (E2, F2α, D2, I2, and thromboxane A2) by specific PG synthase according to the requirements of the organism [17,18,19]. In this way, in intestinal epithelial cells (IECs) the main function of PGE2 is to ensure and regulate the secretion of mucus to protect the gastric mucosa and maintain its integrity at the baseline [20]. In this process, we can say that the key enzymes are the isoforms of COX: COX-1 and COX-2. COX-1, designated as constitutive, is responsible for producing maintenance prostaglandins while COX-2, called inducible, is positively regulated during inflammatory processes [21]. In this context, non-selective NSAIDs such as indomethacin (IND) increase the production of reactive oxygen species (ROS) [22] and block PGE2 synthesis by the inhibition in the gastrointestinal tract of COX-1 and COX-2, resulting in a marked decrease in gastric mucus, bicarbonate production, mucosal blood flow, and in its cytoprotective effects. Therefore, non-selective NSAIDs increase the susceptibility of the gastric mucosa to injury [23].

Spirulina (*Arthrospira maxima*) (SP) is a species of cyanobacteria that has been used by humans since ancient times as an aliment for its high nutritional value and high protein content of up to 70%, *w*/*w* [24]. In the last decades, SP has aroused the interest of several researchers due to its great antioxidant capacity given for valuable biologically active molecules that it owns as tocopherols, carotenoids, ascorbic acid, glutathione, chlorophyll derivatives, and phycobiliproteins (PBPs) between others [25,26]. PBPs are protein–pigment complexes described as accessory photosynthetic pigments of cyanobacteria and red algae. Their structure consists of phycobilins (tetrapyrrole prosthetic groups) covalently attached to these water-soluble proteins and are classified into three main groups: phycoerythrin (PE), allophycocyanin (APC), and phycocyanin (PC) [27]. In multiple investigations, these photosynthetic pigments have shown interesting pharmacological effects as neuroprotective, anticancerous, anti-inflammatory, hepatoprotective, hypocholesterolemic, antioxidant, and gastroprotective by preventing the development and accelerating the healing of gastric ulcers induced by ethanol or stress [28,29,30,31]. Hence, since there is no information on the gastroprotective effect of PBPs in a gastric ulcer induced by non-selective NSAIDs of COX-2, this work sough to bridge this gap by studying the effect of PBPs to reduce gastric damage derived from the use of IND. Hence, the present research evaluated the gastroprotective effect of an aqueous extract (PhyEx) of SP rich in PBPs on IND-induced gastric ulcers in rats. As such, the histopathological damage at the level of the stomach mucosa was evaluated as well as some of the main antioxidant and oxidative stress markers.

## 2. Results

### 2.1. Evaluation of Phycobiliprotein Content and Purity of PhyEx

Considering the amount of SP used for each extraction cycle (10 g), as well as the amount of lyophilized PBPs obtained on average in each cycle (≈2.077 g), a yield of 20.77% was obtained. Therefore, we can define that the amount of total PBPs present in the SP used is approximately 20%. In addition, as indeed it was reported in a previous publication, the PBPs purity was 0.86 and 0.81 to c-phycocyanin (C-PC) and APC, respectively. Moreover, its individual content was of 0.40 and 0.56 mg/mL [32].

### 2.2. Antiulcerogenic Activity of the Aqueous Extract

Figure 1 shows the main anatomical regions of rat stomach saline-treated and opened along or paramedian to the greater curvature. The main areas we can observe are the fundus (aglandular region) and corpus (glandular region) separated by a dotted red line (Limiting ridge) that surrounds the lower part of the esophagus grove (dotted purple line) and that is located above the region of the antrum (dotted blue line). The antrum in turn is limited in its lower part by the pyloric sphincter (dotted black line).

Figure 2a shows that pretreatment with the vehicle in the control group does not generate macroscopic lesions or changes in the continuity of the mucosa, ulcer index (UI) 0, while stomachs treated with IND 40 mg/kg revealed multiple oval ulcerations with hemorrhage (Figure 2b) (UI, 2.81 ± 0.33), as well as the presence of clots at the base of the lesions. Meanwhile, stomachs of groups that received cimetidine (CTD) 100 mg/kg plus IND 40 mg/kg (UI, 0.15 ± 0.01) or 100, 200, and 400 mg/kg of PhyEx plus IND 40 mg/kg (UI, 2.07 ± 0.22, 2.15 ± 0.23, 0.92 ± 0.17, respectively) showed a significant decrease in the gastric lesions induced by IND, thus better preserving the mucus layer at the highest dose (*p* < 0.05) (Figure 2c–f, Table 1). Concerning the PBPs, doses of 100, 200, and 400 mg/kg of PhyEx offered significant protection against IND-induced ulcers of 26.47, 23.57, and 67.19%, respectively (*p* < 0.01). On another hand, a better protection percentage was offered by treatment with CTD 100 mg/kg (94.59%) compared with the IND 40 mg/kg group (0%) (Table 1).

### 2.3. Histopathology

Microscopic analysis of the gastric mucosa (H and E staining) of the control group revealed an intact mucosa, as well as glands with the regular arrangement (Figure 3a,b). The IND 40 mg/kg group showed several lesions of a necrotic type with notable disruption of the epithelium, naked nuclei, and reactive cellular changes in the cytoplasm of glandular cells with a granular appearance. Moreover, medium edema was observed in the submucosa with leukocyte infiltrate (Figure 3c,d). Comparatively, the pretreated group with PhyEx 400 mg/kg presented better protection of the gastric mucosa against IND 40 mg/kg, as evidenced by a reduction in total areas lesioned, edema in the submucosa, remnant presence of mucus at the edges of the ulcer, and a lower degree of necrosis (Figure 3g,h). The best PP was observed at the highest dose; finally, the CTD 100 mg/kg group was characterized by slight damage to the mucosal epithelium, minimal or no edema, and little necrosis (Figure 3e,f).

### 2.4. Antioxidant Enzymes and Lipoperoxidation

To investigate the participation of antioxidant defenses in PhyEx gastroprotection against IND-induced damage, the activity of the main antioxidant enzymes (SOD, CAT, and GPx) as well as the content of MDA were evaluated (Figure 4). In the IND 40 mg/kg group, a significant depletion in the gastric activity of SOD, CAT, and GPx enzymes was observed, as well as an increase in the MDA levels compared to the control group. Alternatively, pretreatment with PhyEx was able to significantly restore the diminished activity of these same antioxidant enzymes and decreased the levels of MDA (*p* < 0.05) in comparison with IND 40 mg/kg, achieving similar values to those of the control group. Similarly, pretreatment with the CTD 100 mg/kg group helped to significantly restore the low activity of the SOD, CAT, and GPx enzymes and reduced MDA to levels close to normal.

## 3. Discussion

For several years it has been known that gastric mucous protection can be impaired by NSAID administration by inhibiting its production and simultaneously promoting the production and accumulation of ROS [33]. Additionally, when NSAID use has been prolonged, NSAIDs can lead to gastroduodenal disorders including severe gastric or duodenal ulcers [34].

Throughout human history, plants and their derivatives have been used around the world to treat several ailments or diseases. Nowadays, a considerable part of scientific research has focused on the scrutiny of herbal medicine as an alternative to commercially synthetic drugs available to treat multiple diseases including GU [35]. In this sense, plants with notable antioxidant activity such as *Rubus crataegifolius* [36], *Ziziphus jujube* [37], *Cuphea ignea* [38], and Spirulina *Arthrospira platensis* [39] have proved to be effective to protect against the damage in different ulcer models of gastric ulcers. Therefore, in this study, the protective effect of pretreatment with the PBPs of SP in a gastric ulcer model using IND 40 mg/kg as an ulcerogenic agent in the rat stomach was determined. Additionally, the effect of PhyEx was evaluated on the main antioxidant (SOD, CAT, and GPx) and prooxidants (MDA) markers as well as in the changes in the gastric epithelium. Since in our experiment, the purity of obtained PC and APC in the PhyEx was considered good, a purity of 0.7 is accepted as food grade [40], so it was possible to continue with the rest of the experiments.

In the IND group, the significant increase in MDA levels was accompanied by significant ulceration, and a significant decrease in the activity of the SOD, CAT, and GPx enzymes compared to the control was observed. It can lead to the accumulation of ROS, the concentration of MDA, and the increase in susceptibility to damage of gastric mucosa. Alternatively, treatment with PhyEx at 400 mg/kg increased the activity of SOD, CAT, and GPx, and decreased MDA levels. Hence, to explain the gastroprotective effect of PhyEx, it is deduced that it reduced the lipoperoxidation (MDA levels) and restored the levels of SOD, CAT, and GPx by inhibiting the oxidative damage caused by IND in the gastric mucosa of rat stomachs. As mentioned before, SP contains a wide variety of compounds with the ability to trap ROS, among which PBPs are presumed to be primarily responsible for such activity [41]. These light-harvesting pigments were almost exclusively investigated for their antioxidant potential, although it is known that they have two primary routes to express their antioxidant activity (PE catch ROS through a redox reaction, while, in addition to this same mechanism, PC and APC chelate metal ions) [31], it was theorized that its antioxidant action integrates several mechanisms associated with the side chains of constitutive amino acids, as well as variations in the distribution of the amino acids in the surface of PBPs [42,43]. Damage to the gastric mucosa observed in IND 40 mg/kg can be caused by ROS, lipoperoxidation, and a decreased level of antioxidant defense. Several reports indicate that consumption of IND generates ROS in normal gastric epithelial cells by promoting the uncoupling of oxidative phosphorylation [22] and the inhibition of the electron transport chain [44], producing the incomplete reduction of molecular oxygen. Once produced ROS, its accumulation leads to damage in tissues by attacking three main molecules, namely DNA, proteins, and lipids, and disrupting cellular signaling [45]. Additionally, the damage of mitochondrial aconitase releases iron from its clusters, which reacts with ROS to produce various free radicals. In this way, numerous authors have described oxidative stress as an essential factor in the pathogenesis of GU by excessive NSAID consumption [33,46,47,48].

Consistent with such reports, in our study the administration of IND 40 mg/kg produced marked damage in the gastric epithelium of rat stomachs, evidenced by mucous loss and the presence of elongated lesions with intense hemorrhaging and hyperemia. In contrast, the pre-treatment of rats with PhyEx drastically attenuated the gastric damage, allowing the recuperation of mucosa integrity and indicating that PhyEx has a protective effect against the ulcerative lesions induced by IND due to its strong antioxidant activity, the ability of PE to protect against cellular injury [49,50], and the potent wound-healing effect in vivo and in vitro of PC to induce cell regeneration [51,52].

On another hand, the macroscopic analysis showed that IND induced considerable gastric damage by comparison to the control. IND 40 mg/kg induced mucosal damage characterized by necrosis, edema, and leukocyte infiltration in the submucosa. The treatment with PhyEx significantly reduced gastric damage induced by IND, evidenced by the decrease of edema in the submucosa as well as the presence of mucus in lesioned areas.

In this context, PG plays a fundamental role in the maintenance of physiological processes such as blood flow in the gastric mucosa, angiogenesis, mucus secretion, bicarbonate, and participation in the epithelial restitution process [53]. Additionally, PGs can inhibit mast cell activation, as well as the adhesion of platelets and leukocytes to the vascular endothelium [54]. As previously mentioned, PGs are synthesized by COX-1 and COX-2 (COX-2 is expressed mainly in pathological processes). The expression of COX-2 increases in the case of damage to the gastric mucosa [55]. However, although a COX-dependent protective mechanism of PhyEx was not explored in the present study, it is known that C-PC is a natural COX-2 inhibitor [56] which decreases the synthesis of PGE2 which, depending on the context, actively participates in the inflammatory response. This allows us to assume that the decrease in the inflammatory infiltrate and edema in the group treated with PhyEx could have been influenced by a decrease in PGE2 levels derived from possible COX-2 inhibition. Since only the anti-ulcerogenic and antioxidant effects of PBPs were evaluated against IND in this study, it is interesting to consider future research to study the underlying specific mechanisms of the gastroprotective effect of PBPs because GU IND induction goes beyond oxidative stress. IND administration non-selectively inhibits both COX isoforms, which decreases mucosal defenses by inhibiting COX-1 (decreases bicarbonate and mucus secretion, and mucosal blood flow) and alters repair processes by inhibiting COX2 (increases leukocyte adherence and decreases epithelial proliferation, angiogenesis, and growth factor release). This in turn leads to inhibition of PGE2 synthesis, decreased mucosal hydrophobic surface and epithelial barriers, and increased mitochondrial dysfunction, apoptosis, and mucosal injury [57,58]. Different authors agree that the main possible mechanisms of IND to develop GU include (but are not limited to) mitochondrial damage by ROS, apoptosis, and the inhibition of PGE2 synthesis [59]. Therefore, there are several specific points in this proposal where PBPs could act to generate their anti-ulcerogenic effect (in addition to its already proven antioxidant effect). It would be interesting to determine whether PBPs prevent COX-1 inhibition in the IND-induced GU model and thus damage mucosal defenses. Furthermore, it would be interesting to evaluate the presence of PGs and how they are affected by PBPs (in particular PGE2) and study whether PBPs, such as SP, selectively inhibit COX-2 and thus prevent alterations in epithelial cell renewal, local micro-ischemia, cellular infiltration, mitochondrial damage. Finally, it would be interesting to evaluate the effect of PBPs on inflammation and necrosis derived from GU development (Figure 5).

## 4. Materials and Methods

### 4.1. Preparation of the Aqueous Extract Rich in Phycobiliprotein

PhyEx was prepared from a suspension of powdered SP of batch number 10-2021 (donated from Alimentos Esenciales para la Humanidad S.A. de C.V., Mexico City, Mexico) in phosphate buffer 20 mmol/L (pH 7.4) (10 g of SP in 40 mL buffer). The suspension was subjected to three consecutive cycles of freezing (−70 °C for at least 2 h) and thawing (37 °C for 30 min). In dark conditions the heated suspension was centrifuged for 30 min at 9 G to discard the biomass. The supernatant was acquired and separated in a new clean centrifugal tube; this step was performed twice to obtain the PBP extract. The PBP extract was lyophilized and stored at −70 °C. Finally, concentrations of PBPs in the supernatant were calculated from the absorption coefficients measured at 562, 620, and 652 nanometers (nm). The procedure is described in detail in our previous publication [32].

### 4.2. Animals and Housing

In total, 60 male Wistar rats of 10 weeks of age (200-250 g) were obtained from the bioterium of the Universidad Autónoma del Estado de Hidalgo (UAEH). A total of 36 rats divided into 6 groups (*n* = 6) were used to evaluate the antiulcerogenic activity, and another 24 rats were distributed into 4 groups (*n* = 6) to evaluate the activity of the antioxidant enzymes. All animals were maintained in cages with raised floors and wide mesh (to prevent coprophagy) in a room under standard conditions of temperature (22 ± 1 °C), relative humidity (55–60%), and a 12/12 h light-dark cycle (lights on at 7:00 a.m.). They had access to food (Rodent Lab Chow 5001, Purina, St. Louis, MO, USA) and purified water ad libitum throughout the experiment. Before inducing ulcers with indomethacin, rats fasted for 19 h. After each experiment, the animals were euthanized in a carbon dioxide euthanasia chamber. The current protocol was approved by the Comité de Ética en Investigación of the Escuela Nacional Ciencias Biológicas (CEI-ENCB082016, Mexico City, Mexico). All procedures and handling of the animals occured in accordance with the Mexican Official Regulation (NOM ZOO–062-200-1999) entitled “Technical Specifications for Production, Care, and Use of Laboratory Animals”.

### 4.3. Drugs and Chemicals

Cimetidine (CTD) and indomethacin (IND) were acquired from Sigma-Aldrich (St. Louis, MO, USA). Thiobarbituric acid (TBA), trichloroacetic acid (TCA), and formaldehyde (FD) were purchased from Merck (Darmstadt, Germany). SOD and GPx were obtained from Randox (Crumlin, Country Antrim, UK). Other reagents and solvents were procured from local sources and were of analytical grade.

### 4.4. Antiulcerogenic Activity of the Aqueous Extract

The experiments in this model were performed according to the method proposed by Santin [60] with modifications: animals were assigned an identification number by coat staining and randomly distributed in the six groups of which our experiment consisted (*n* = 6): (a) control (vehicle, purified water-tween 80, 1%); (b) IND 40 mg/kg; (c) cimetidine (CTD) 100 mg/kg; and (d), (e), (f) PhyEx at 100, 200, and 400 mg/kg, respectively. Within our experimental design, IND was used as a positive control to study gastric damage produced by a COX-2 non-selective NSAID, while CTD was employed as a protective drug for IND-induced gastric damage against which to compare the gastroprotective effect of PBPs. The doses of the substances used in this experiment were taken from those reported in the literature to produce gastric ulcers with IND [61], to protect against gastric damage with CTD [62], and from previous studies in our laboratory in which PBPs showed good antioxidant and anti-inflammatory effects between doses 100 to 400 mg/kg [32].

All treatments were administered by gavage once a day for 8 consecutive days before the induction of gastric ulcer with IND 40 mg/kg, except for group (b) which only received the vehicle during that period [32,61]. On day 8 of the experiment, the corresponding treatment was administered to each group normally and 1 h afterthe gastric lesion was induced and administered by gavage one dose of IND 40 mg/kg [63] to each group, with the exception of the group (a) that only received the vehicle. All treatments were administered in a constant volume of 10 mL/kg.

Four hours after ulcer-inducing, animals were sacrificed in a CO_2_ chamber to obtain the stomachs that were fixed in formaldehyde (FD) 4%. After 10 min, the stomachs were opened over the greater curvature and rinsed with physiological saline solution to remove the blood clots. Next, each gastric sample was placed on a slide and the gastric damage area (mm^2^) was determined by “ImageJ” image processing software. The ulcer index (UI) for each stomach was calculated using the formula:$$\mathrm{UI}=(\mathrm{TAML}\left({\mathrm{mm}}^{2}\right))(\mathrm{TMA}\left({\mathrm{mm}}^{2}\right))$$
where TMA is the total mucosal area while TAML is the total area of mucosal lesion of each rat [64].

The protection percentage (PP) was calculated using the following formula:$$\mathrm{UI}=\frac{\left(\mathrm{TAML}\left({\mathrm{mm}}^{2}\right))(100\right)}{\mathrm{TMA}\left({\mathrm{mm}}^{2}\right)}$$
$$\mathrm{PP}=\frac{\left(\mathrm{UI}\mathrm{control}-\mathrm{UI}\mathrm{treated}\right)}{\left(\mathrm{UI}\mathrm{control}\right)\left(100\right)}$$
where “UI control” is the UI determinate for IND 40 mg/kg group and “UI treated” is the UI of treated groups (d–f) [65]. For other tests, the PhyEx concentration with the least UI was selected.

### 4.5. Enzymatic Activity

#### 4.5.1. Stomach Tissue Preparation

In total, 24 male Wistar rats were distributed in the four groups of which our experiment consisted (*n* = 6): (a) control (vehicle, purified water-tween 80, 1%); (b) IND 40 mg/kg; (c) cimetidine (CTD) 100 mg/kg; and (d) PhyEx at 200 mg/kg. The administration regimens were the same as those of the previous experiment.

After sacrifice, stomachs were obtained and cut along the greater curvature and rinsed with cold phosphate buffer (PBS) pH 7.4. A portion of 0.5 g was approximately cut into small pieces and placed in 4.5 mL of cold PBS to be homogenized on an ice bath with an Ultra-turrax homogenizer (T18, IKA, Staufen im Breisgau, Germany) and a Polytron handheld homogenizer (Newtown, CT, USA). Homogenate tissues were centrifuged (12 min at 12,000 rpm, 4 °C) and the supernatants were aliquoted (500 μL c/u) and stored at −20 °C until their use [32].

#### 4.5.2. Glutathione Peroxidase Activity

Glutathione peroxidase activity (GPx) in the supernatant of gastric tissue was determined using a commercial kit Ransel RS504 according to the description by Paglia and Valentine [66]. Posterior to the reaction with the sample, the absorbance was measured after 1 and 2 min at λ of 340 nm. Enzyme activity was directly proportional to the rate of change.

#### 4.5.3. Superoxide Dismutase Activity

Superoxide dismutase activity (SOD) in the supernatant of gastric tissue was determined using a commercial Ransod SD125 kit according to the protocol described by McCord and Fridovich [67]. In this procedure, the SOD activity was determined as the degree of inhibition of this reaction; absorbance was measured at λ of 505 nm.

#### 4.5.4. Catalase Activity

The catalase activity (CAT) in the supernatant of gastric tissue was evaluated by tracking the rate of decomposition of H_2_O_2_ in the presence of CAT at λ of 240 nm [68].

### 4.6. Total Proteins

The total content of protein in supernatants was determined through the Bradford method using bovine serum albumin as a standard [69]. The blue protein dye was detected at 595 nm.

### 4.7. Lipoperoxidation Assessment

To evaluate the levels of lipoperoxidation in the samples, the content of malondialdehyde (MDA) was determined in the supernatants by thiobarbituric acid reactive substances (TBARS) assay, as described by Esterbauer and Cheeseman [70]. The assay consisted of 0.5 mL of gastric mucosal homogenates and 1.0 mL of reactive mixture: 0.375% of TBA and 15% of trichloroacetic acid (TCA) in 0.20 N of HCl. After incubation for 15 min in boiling water, the samples were cooled and centrifuged at 1000 rpm for 10 min at 4 °C. The absorbance of the supernatant was measured at λ of 532 nm and the concentration of MDA was calculated with an extinction coefficient of 156,000 M^−1^ cm^−1^.

### 4.8. Histopathological Examination

After the stomachs were fixed in 10% formalin solution for 24 h, they were dehydrated in ascending concentrations of alcohol solutions and subsequently they were included in paraffin. Then, slides of the stomach (4–5 μm of thickness) were prepared and samples were stained with hematoxylin and eosin (H and E) and analyzed under an optical microscope at 20× and 40× for pathological changes such as vasocongestion, necrosis, eosinophilic infiltration, glandular damage, and edema. All slides were photographed using a Zeiss Axiophot microscope (Thornwood, NY, USA) [32].

### 4.9. Statistical Analysis

Statistical analysis was carried out with SigmaPlot version 14.0 (Systat Software, San Jose, CA, USA). All data are expressed as the mean ± standard error of the mean (SEM). One-way analysis of variance (ANOVA) and the post hoc Student–Newman–Keuls test were used to compare the treated groups; statistical significance was considered at *p* < 0.05.

## 5. Conclusions

The results suggest that the protection against GU may be due to the antioxidant properties of PhyEx that activated antioxidant mechanisms (SOD, CAT, and GPx) to decrease lipid peroxidation (MDA) and reduce the inflammatory response. However, further studies are needed to explore and clarify the mechanisms underlying the gastroprotective effect shown by PhyEx.

## Figures and Tables

**Figure 1 plants-12-01586-f001:**
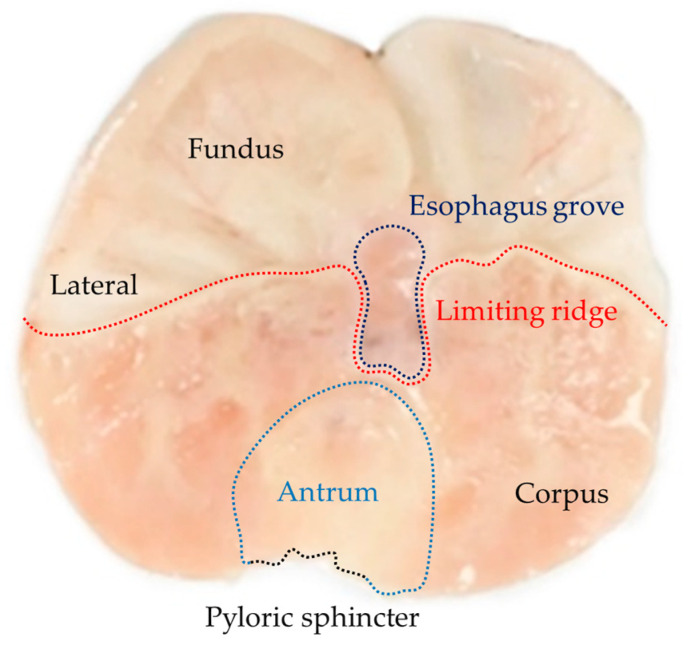
Rat stomach opened at the greater curvature showing the main anatomical parts: the fundus, esophagus grove (purple), limiting ridge (red), antrum (blue), and corpus pyloric sphincter (black).

**Figure 2 plants-12-01586-f002:**
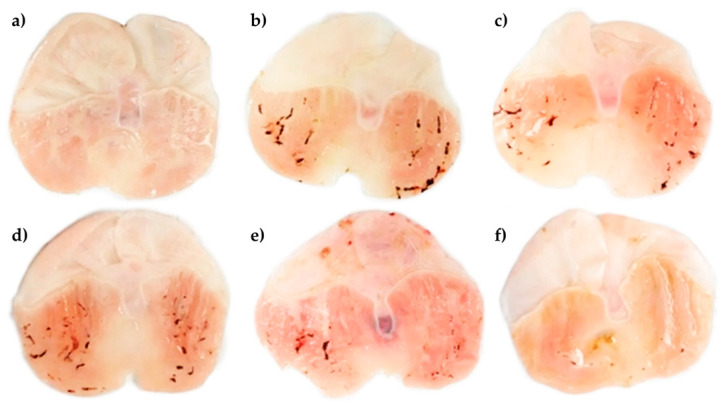
Macroscopic appearance of the gastric ulcer area of IND-induced in rat stomachs. (**a**) Control, normal appearance of the corpus region without gastric ulcers; (**b**) IND 40 mg/kg, necrotic ulcerations throughout the corpus; (**c**) CTD 100 mg/kg + IND 40 mg/kg, smaller number and size of necrotic ulcerations on the corpus; (**d**) PhyEx 100 mg/kg + IND 40 mg/kg, less protective effect against the development of gastric ulcers on the corpus; (**e**) PhyEx 200 mg/kg + IND 40 mg/kg, intermediate protective effect against the development of gastric ulcers on the corpus; (**f**) PhyEx 400 mg/kg IND 40 mg/kg, better protective effect against the development of gastric ulcers on the corpus.

**Figure 3 plants-12-01586-f003:**
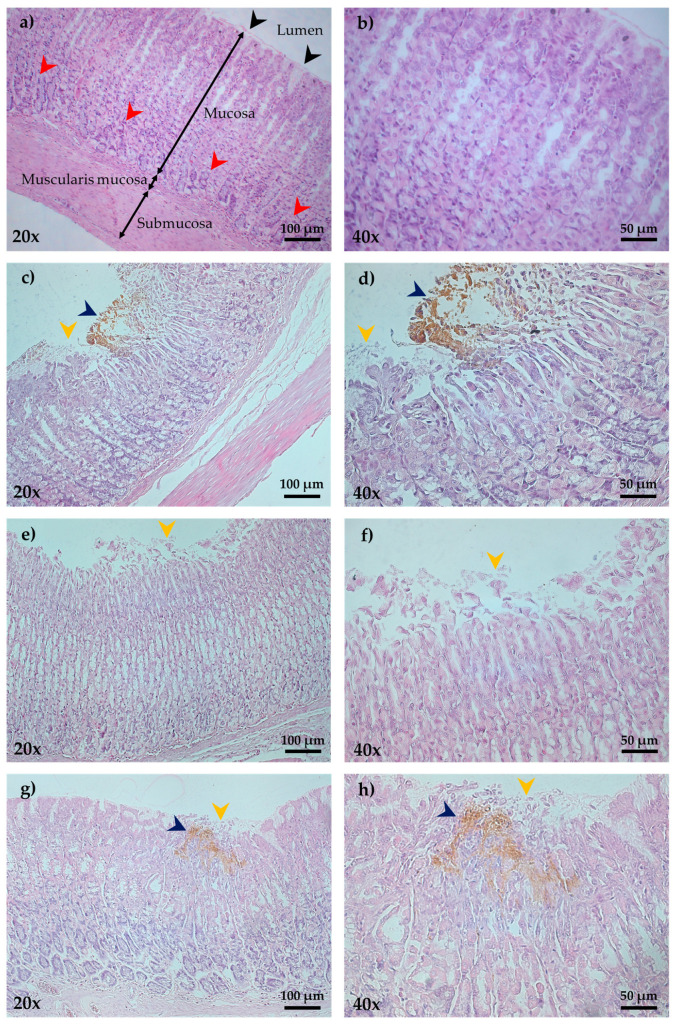
H and E staining of gastric mucosa of rats in the IND-induced gastric ulcer model (magnification at 20× and 40×). (**a**,**b**) Control, shows normal aspect of the lumen, mucosa, muscularis mucosa, and submucosa strata; (**c,d**) IND 40 mg/kg, shows severe hemorrhagic necrosis with leukocyte infiltrate (dark blue arrowheads) and erosion of the surface epithelial cells (orange arrowheads); (**e,f**) CTD 100 mg/kg + IND 40 mg/kg, shows erosion of the surface epithelial cells (orange arrowheads); (**g,h**) PhyEx 400 mg/kg + IND 40 mg/kg, shows moderate hemorrhagic necrosis (dark blue arrowheads) and erosion of the surface epithelial cells (orange arrowheads). The region of the epithelium is indicated by black arrowheads while region of the lamina propria is indicated by red arrowheads A).

**Figure 4 plants-12-01586-f004:**
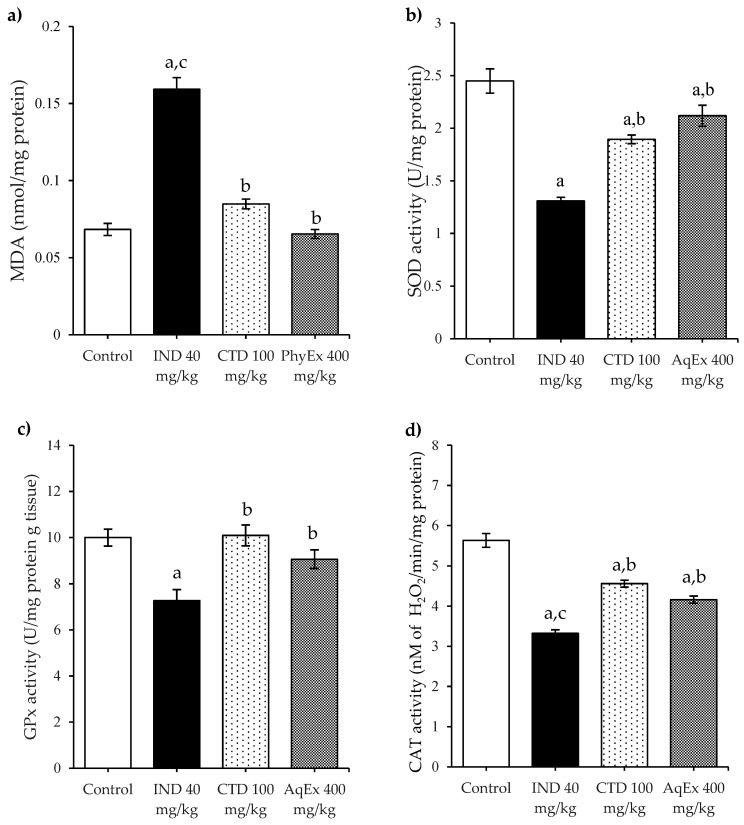
Effects on the content of MDA (**a**), and the activity of SOD (**b**), GPx (**c**), and CAT (**d**) enzymes, after pretreatments with PhyEx and CTD in the gastric ulcer model induced by IND. PhyEx, aqueous extract; CTD, cimetidine; IND, indomethacin. Data are expressed as the mean ± SEM (*n* = 6). One-way ANOVA *post hoc* Student–Newman–Keuls showed a significant difference (*p* < 0.01). Literals indicate differences vs. groups: ^a^ Control; ^b^ IND 40 mg/kg; ^c^ PhyEx 400 mg/kg.

**Figure 5 plants-12-01586-f005:**
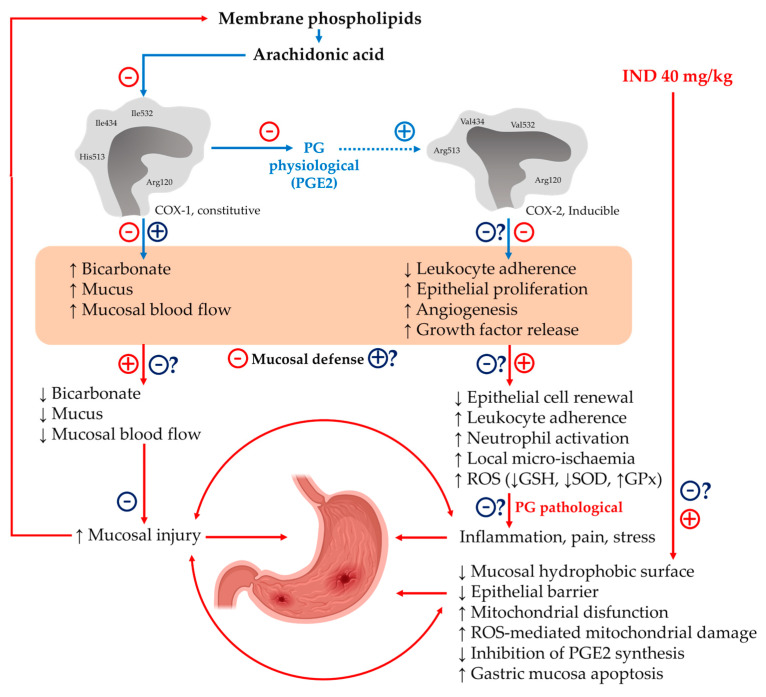
Critical points in the development of an indomethacin-induced gastric ulcer. The main steps in the development of IND-induced GU (red symbols and arrows) in which PBPs could act to exert their antiulcerogenic effects (dark-blue symbols) within the normal maintenance of gastric mucosa (blue symbols and arrows) are shown. Arachidonic acid is released from the phospholipids of lipid membranes. It is then oxidized by COX-1 to various prostaglandins, mainly PEG2, to regulate and maintain mucosal defenses and thereby ensure the integrity of the membranes. In this scenario, the administration of IND blocks the synthesis of PGE2 thus decreasing the secretion of protective substances such as gastric mucus, bicarbonate, and mucosal blood flow. This increases ROS production, mitochondrial damage, and apoptosis; generates painful inflammatory processes; and increases susceptibility to mucosal damage. Likewise, PBPs could reduce the harmful effects derived from COX-1 and COX-2 inhibition, thereby reducing mitochondrial damage and apoptotic processes and increasing angiogenesis and gastric epithelial renewal, among other things.

**Table 1 plants-12-01586-t001:** Effect of PhyEx and CTD on ulcer parameters in rats with IND-induced ulcers.

Groups	Treatments (mg/kg)	Ulcer Index (mm^2^)	Protection Percentage (%)
I	Control (vehicle)	0.0 ± 0.00	0.0
II	Vehicle + IND 40	2.81 ± 0.33	0.0
III	CTD 100 + IND 40	0.15 ± 0.01	94.59 ^a^
IV	PhyEx 100 + IND 40	2.07 ± 0.22	26.47 ^a^
V	PhyEx 200 + IND 40	2.15 ± 0.23	23.57 ^a^
VI	PhyEx 400 + IND 40	0.92 ± 0.17	67.19 ^a^

The values are expressed as the mean ± SEM; *n* = 6; PhyEx, aqueous extract; CTD, cimetidine; IND, indomethacin. Significant difference (*p* < 0.05) vs. ^a^ Vehicle + IND (40 mg/kg).

## Data Availability

The data presented in this study are available on request from the corresponding author.

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
