# Peer review of "Protective Effect of the Phycobiliproteins from Arthrospira maxima on Indomethacin-Induced Gastric Ulcer in a Rat Model"

_plants, 2023, doi:10.3390/plants12081586_

Round 1

Reviewer 1 Report

This manuscript describes the investigation on the protective effect of the phycobiliproteins from Arthrospira maxima on indomethacin-induced GU (gastric ulcer) in a mouse model. This topic is important in that there are more than 8 million of the global population affected by GU. I think the results are interesting and have the potential to lead to more investigations in this area. The results did suggest the reduction of inflammatory although the mechanism of this reduction needs more investigation. Thus, I recommend publishing after some minor revisions.

1.      Page 3, line 142, “Figure 10” was mentioned but there is no figure 10 in this manuscript. This should be Figure 2? Please check the manuscript carefully to ensure that all the numbers are consistent in the text and Figures and tables.

2.      There is “Figure 3” but there is no mention of “Figure 3” in the text. No reference to “Figure 3” in the paragraphs leading up to this figure as well as the paragraphs after this figure.

Author Response

We appreciate the time and effort dedicated to providing your valuable comments to improve the quality of this manuscript. We have attended to your comments and suggestions. In this regard, a point-by-point response is attached. We hope you find our responses satisfactory. 

Comment 1: Page 3, line 142, “Figure 10” was mentioned but there is no figure 10 in this manuscript. This should be Figure 2? Please check the manuscript carefully to ensure that all the numbers are consistent in the text and Figures and tables.

Response 1: Thank you for your insightful observation. Instead of figure 10 it should read figure 4 to indicate the group of graphs showing the activities mentioned. The entire manuscript has been reviewed to verify that the figure numbers mentioned in the text match the figure numbering. All the correction has been made (lines 155 and 208).

Comment 2: There is “Figure 3” but there is no mention of “Figure 3” in the text. No reference to “Figure 3” in the paragraphs leading up to this figure as well as the paragraphs after this figure.

Response 2: Thank you for pointing this out. Your observation is related to your previous comment 1; the missing mention of figure 3 is because it was mistakenly written as figure 10. And since one more figure was added, the previous figure 3 is now figure 4 (lines 155 and 208).

Reviewer 2 Report

The research topic in the manuscript is interesting, but I have many objections to the manuscript itself.

1.       In section 2.1, the Authors mention that they refer to results from a previous publication, but do not indicate which publication they are referring to.

2.       In section 2.2, the Authors describe the results as if there were groups exposed only to CTD or PhyEx.

3.       In section 2.3, the Authors indicate that there are microscopic images for groups exposed to PHyEx of 100 and 200 mg/kg, while there are no such figures.

4.       The authors use different forms of the abbreviation GPx.

5.       The caption of Figure 1 suggests that there were groups exposed only to CTD or PhyEx.

6.       The caption of Figure 3 shows that the Authors examined MDA activity, while in the text of the manuscript the Authors report that they examined MDA levels.

7.       The presentation of statistical significance in Figure 3 is difficult to understand.

8.       In section 4.3, the Authors indicate that they used SOD and GPx, while later in the chapter they write that they studied the activity of these enzymes in supernatants.

9.       The Authors point out that the studies were undertaken in 33 rats, and in section 4.4 They write that the animals were divided into 6 groups of 6 rats each (36).

10.   In section 4.4, the Authors indicate that group a was the control group, and further indicate that group b received only the vehicle.

Author Response

We appreciate the time and effort dedicated to providing your valuable comments to improve the quality of this manuscript. We have attended to your comments and suggestions. In this regard, a point-by-point response is attached. We hope you find our responses satisfactory. 

Comments from reviewer 2

Comment 1: In section 2.1, the Authors mention that they refer to results from a previous publication, but do not indicate which publication they are referring to.

Response 1: Thank you for pointing this out, the quote from the above publication has been added on the corresponding section (line 118).

Comment 2: In section 2.2, the Authors describe the results as if there were groups exposed only to CTD or PhyEx.

Response 2: We agree with the reviewer’s assessment. Accordingly, throughout section 2.2 we have specified the treatments administered “plus IND 40 mg/kg” (lines 130-133). 

Comment 3: In section 2.3, the Authors indicate that there are microscopic images for groups exposed to PHyEx of 100 and 200 mg/kg, while there are no such figures.

Response 3: Thank you for pointing this out. Indeed, the figure 2 only shows histological images of the PhyEx group with the best protective effect (PhyEx 400 mg/kg). So, the text has been corrected (line 146).

Comment 4: The authors use different forms of the abbreviation GPx.

Response 4: You are right, after reviewing our manuscript we found that we had variants with upper and lower case (Gpx, GPx, GPX). We made changes in the text to use only "GPx".

Comment 5: The caption of Figure 1 suggests that there were groups exposed only to CTD or PhyEx.

Response 5: Thank you for pointing this out. You are right, for lack of information in the caption of Figure 1 (now figure 2), seems that the groups only received CTD or different doses of PhyEx. Changes have been made to specify that such groups were treated with indomethacin 40 mg/kg (lines 176-181).

Comment 6: The caption of Figure 3 shows that the Authors examined MDA activity, while in the text of the manuscript the Authors report that they examined MDA levels.

Response 6: Your comment is correct, the caption of figure 3 (now figure 4) was written to include all 4 markers MDA, SOD, GPx and CAT. However, this can lead to some inconsistencies with the text such as the one you point out. Therefore, the caption of Figure 4 has been updated to be more accurate “Effects on the content of MDA (a), and activity of SOD (b), GPx (c), and CAT (d) enzymes…” (line 208).

Comment 7: The presentation of statistical significance in Figure 3 is difficult to understand.

Response 7: We appreciate the reviewer’s feedback. We have modified the presentation of statistical significance in Figure 4 (before figure 3) caption to specify that the literals show differences against a given group/treatment. We are confident that the significance is now better understood (line 212).

Comment 8: In section 4.3, the Authors indicate that they used SOD and GPx, while later in the chapter they write that they studied the activity of these enzymes in supernatants

Response 8: That is correct, tissue samples were taken and according to the cited methodologies; after that, collected samples were homogenized and centrifuged. Subsequently, enzyme determinations were performed on the supernatants following the instructions of the enzyme kits used.

Comment 9: The Authors point out that the studies were undertaken in 33 rats, and in section 4.4 They write that the animals were divided into 6 groups of 6 rats each (36).

Response 9: Thank you for your kind comment. We agree with you; in fact, 6 groups with 6 animals each were used to study the antiulcerogenic activity of the aqueous extract, so the total number of animals used was 36. In addition, 24 more rats were used for ezymatic determination and histological analysis. Thus, the total number of animals in this study was 60 rats. The correction has been made in the indicated section (lines 336-340).

Comment 10: In section 4.4, the Authors indicate that group a was the control group, and further indicate that group b received only the vehicle.

Response 10: Thank you for pointing this out. We believe that there was a confusion because the experiment consists of two types of treatment administration; that of the treatment prior to the induction of gastric ulcer; and that of the induction of gastric ulcer. In the first, all treatments except for control and IND were administered to protect from future damage (CTD or the three doses of PhyEx). Since we cannot administer IND at that period to IND group, the group IND 40 mg/kg received only the vehicle (so that it also has the stress of gavage). And on the day of gastric ulcer induction, IND was administered to all groups (except control). We have rewritten some sentences of section 4.4 and we believe that now the process of pretreatment and ulcer induction is clearer (lines 370-376).

Reviewer 3 Report

The study highlighted the protective effects of S. maxima aqueous extract against oxidative damage in an animal model of gastric ulcer

1. State the research gap considering S. maxima has been tested on the recovery of gastric ulcer- Ref 28-31.

2. State the novelty of the study

3. Provide the details of the preparation for PhyEx (aqueous extract) and PBP. What is the amount of raw material needed, herbarium no (if any), batch no, and the ratio of raw material to water.

4. 33 rats were obtained for experiments. There were 6 rats in a group, 4 groups in total = 24 rats. Clarify the discrepancy.

5. 4.1 Drugs and chemicals

6. State the role of IND and cimetidine in the Methodology. How did get its single optimum dose for this study? Similarly, justify the selected concentration of PhyEx - 100, 200, and 400 mg/kg used in this study. Include a supplementary data for pilot test

7.Please check for grammatical error and sentence construction for line 292-297

8. Pre-treatment was performed for 8 days. State the reference(s)

9. In 4.5.1, did the authors use a second batch of rats? If so, the total number of rats must be clearly indicated in 4.2. How were these rats distributed according to experimental designs?

10. Include reference(s) for each protocol and bioassay

11. The study did not test on PBP. How will you define the amount of SP for PhyEx of SP rich in PBPs

12. The % of protection in PhyEx groups range from 23-67%. A concentration higher than 400 mg/kg should be tested. How were the concentrations of PhyEx determined. Did the authors observe signs of toxicity on a pilot study using concentrations above 400 mg/kg?

13. Figure 1(f). Alphabet (f) appears red.

14. Figure 1. Label the anatomical parts of the stomach

15. Label histological features in Figure 2 - epithelium, lamina propria, smooth muscle fibres and muscularis mucosae. Explain the brown stain.

16. Orientate the photomicrographs in Figure 2 so that the mucosa is located superior to the underlying submucosa. Explain the difference in intensity between A and (B+C+D).

17. Please explain the small alphabets above the bars in Figure 3, eg Asterisk (*) symbol indicates significant difference (p < 0.05; Games-Howell) in viability relative to the negative control. 

18. Histopathological changes stated in 2.3 must be shown in Figure 2

...disruption of the epithelium, naked nuclei, and reactive cellular changes in the cytoplasm of glandular cells, with a granular appearance. Besides, medium edema was observed in the submucosa with leukocyte infiltrate...

19. Why only single dose of 400 mg/kg was tested in histopathological study and quantitative analysis of MDA and endogenous antioxidant assay?

20. Separate this section into two parts and revised the subheadings for each. Omit the word indomethacin

2.4. Antioxidant enzymes and lipoperoxidation indomethacin model

21. State the limitation of the study and possible protective mechanisms , aided by a schematic diagram.

Kindly address to these comments by stating the line number (after changes) in the rebuttal letter. 

Author Response

We appreciate the time and effort providing your valuable comments to improve the quality of this manuscript. We have attended to your comments and suggestions rightly suggested. In this regard, a point-by-point response is attached. We hope you find our responses satisfactory.

Comments from reviewer 3

Comment 1: State the research gap considering S. maxima has been tested on the recovery of gastric ulcer- Ref 28-31.

Response 1: Thank you for this suggestion. We agree and believe that it will help to better visualize the impact of the study, so the changes were made in the corresponding section (lines 103-106).

Comment 2: State the novelty of the study.

Response 2: Agree. We have, accordingly modified the corresponding section to emphasize this point (lines 103-106).

Comment 3: Provide the details of the preparation for PhyEx (aqueous extract) and PBP. What is the amount of raw material needed, herbarium no (if any), batch no, and the ratio of raw material to water.

Response 3: Thank you for your comment. We have added details regarding the phycobiliprotein extraction process in the corresponding section. We believe that now the methodology is clearer (lines 324-331).

Comment 4: 33 rats were obtained for experiments. There were 6 rats in a group, 4 groups in total = 24 rats. Clarify the discrepancy.

Response 4: Thank you so much for catching these glaring and confusing errors, which we have now corrected: 6 groups with 6 animals each were used, so the total number of animals used was 36. In addition, 24 more rats were used for ezymatic determination and histological analysis. Thus, the total number of animals in this study was 60 rats. The correction has been made in the indicated section (lines 336-339).

Comment 5: 4.1 Drugs and chemicals

Response 5: It is not clear to us what your comment on “4.1 Drugs and chemicals” refers to.

Comment 6: State the role of IND and cimetidine in the Methodology. How did get its single optimum dose for this study? Similarly, justify the selected concentration of PhyEx - 100, 200, and 400 mg/kg used in this study. Include a supplementary data for pilot test.

Response 6: Thank you for this suggestion. It was interesting to explore this aspect, the necessary information was added in the methodology. Within our experimental design IND was used as a positive control to study gastric damage produced by a COX-2 non-selective NSAID; while CTD was employed as a protective drug for IND-induced gastric damage against which to compare the gastroprotective effect of PBPs. The doses of the substances used in this experiment were taken from those re-ported in the literature to produce gastric ulcers with IND, to protect against gastric damage with CTD, and from previous studies in our laboratory in which PBPs showed good antioxidant and anti-inflammatory effects between doses 100 to 400 mg/kg (lines 362-369).

Comment 7: Please check for grammatical error and sentence construction for line 292-297.

Response 7: We agree with your comment. We have reviewed our manuscript for grammatical errors. After some corrections (especially in the lines mentioned above), we believe that the document is now clearer (lines 370-376).

Comment 8: Pre-treatment was performed for 8 days. State the reference(s).

Response 8: We agree with your suggestion. The reference on which we rely for pretreatment was added in the corresponding section (line 372).

Comment 9: In 4.5.1, did the authors use a second batch of rats? If so, the total number of rats must be clearly indicated in 4.2. How were these rats distributed according to experimental designs?

Response 9: Thank you for your kind suggestions. The information was added in the corresponding section, we believe that the experimental design is now clearer (lines 336-339).

Comment 10: Include reference(s) for each protocol and bioassay.

Response 10: We agree with your suggestion. The missing references on which we rely for each protocol and bioassay were added in the corresponding section (lines 353 409 and 445).

Comment 11: The study did not test on PBP. How will you define the amount of SP for PhyEx of SP rich in PBPs

Response 11: Thank you for your question. Data derived from the analysis of PBPs obtained from the SP extract are not reported in the present study, because such data were already reported in a previous study. In the current study we are only referencing the final values obtained and providing the reference where the complete data can be consulted. On the other hand, considering the amount of SP used for each extraction cycle (10 g), as well as the amount of lyophilized PBPs obtained on average in each cycle (≈2.077 g), we consider that we had a yield of 20.77 %. Therefore, we can define that the amount of total PBPs present in the SP used is approximately 20 % (lines 113-116).

Comment 12: The % of protection in PhyEx groups range from 23-67%. A concentration higher than 400 mg/kg should be tested. How were the concentrations of PhyEx determined. Did the authors observe signs of toxicity on a pilot study using concentrations above 400 mg/kg?

Response 12: Thank you for your question. We have not tested doses higher than 400 mg/kg because most studies with this type of compounds are done with doses ranging from 10 to 250 mg/kg, and some others with higher doses, 300 or 400 mg/kg. So far, we have not observed signs of toxicity at doses of 400 mg/kg, so we believe that higher doses could be tested. However, when investigating a biological activity we consider that doses higher than 400 mg/kg could be unattractive since other drugs on the market have very good effects at lower doses.

To prepare the doses to be administered, the amount of lyophilized extract was weighed and dissolved in the vehicle under dark conditions to avoid photodegradation of the pigments. For example, if our rat weighed 231 g, to administer a dose of 100 mg/kg of phycobiliproteins, 23.1 mg of the lyophilized PBPs were dissolved in 2.31 mL of the vehicle.

Comment 13: Figure 1(f). Alphabet (f) appears red.

Response 13: Thank you for your great observation. We have changed the red color of the letter f) to black.

Comment 14: Figure 1. Label the anatomical parts of the stomach.

Response 14: We find your recommendation very interesting. Since in the previous figure 1 there was not enough space to show the main anatomical regions adequately, we decided to add a new figure 1 completely dedicated to this purpose (lines 120-125, and 171-173).

Comment 15: Label histological features in Figure 2 - epithelium, lamina propria, smooth muscle fibres and muscularis mucosae. Explain the brown stain.

Response 15: Thank you for your kind comments, we are in complete agreement. The histologic features have been labeled; in image 3A) the main regions are shown, while in the rest of the images the main histologic alterations derived from IND treatment are indicated by arrowheads. We believe that figure 3 is now more appropriate (lines 183-191).

Comment 16: Orientate the photomicrographs in Figure 2 so that the mucosa is located superior to the underlying submucosa. Explain the difference in intensity between A and (B+C+D).

Response 16: Thank you for reminding us of the importance of properly orienting the images within the figures. We have made all the suggested changes, both in figure 3 and in its caption. We are confident that the image is now more accurate (lines 183-191).

Comment 17: Please explain the small alphabets above the bars in Figure 3, eg Asterisk (*) symbol indicates significant difference (p < 0.05; Games-Howell) in viability relative to the negative control.  

Response 17: Thank you for your kind comments. At the end of the figure caption of figure 4 (before figure 3), the meaning of each of the literals that appear on the bars of each graph is already explained: “One-way ANOVA post hoc Student-Newman-Keuls showed a significant difference (p< 0.01). Literals indicate differences vs groups: aControl; bIND 40 mg/kg; cPhyEx 400 mg/kg” (line 212).

Comment 18: Histopathological changes stated in 2.3 must be shown in Figure 2

Response 18: Thank you for pointing this out. We have implemented all your suggestions in the images and captions of figure 3 (line 183-191).

Comment 19: Why only single dose of 400 mg/kg was tested in histopathological study and quantitative analysis of MDA and endogenous antioxidant assay?

Response 19: Thank you for your inquiry. We only tested the 400 mg/kg dose of PBPs because in the previous test, when the anti-ulcerogenic index was determined, we observed that the dose that showed the best protection was the 400 mg/kg dose. Therefore, we did not consider it necessary to evaluate the histological alterations at the low and intermediate doses that showed a partial effect, in order to contribute to a lower use of animals.

Comment 20: Separate this section into two parts and revised the subheadings for each. Omit the word indomethacin: 2.4. Antioxidant enzymes and lipoperoxidation indomethacin model

Response 20: Thank you for this suggestion. We agree to change the section title and omit the word indomethacin (line 152). Regarding the separation of the section into two parts, it would have been interesting to explore this aspect. However, in the case of our study it seems slightly out of scope because the antioxidant enzyme activity and the lipoperoxidation are closely related. Thus, we believe it is more appropriate to have them in a single section so that the reader can look at both sets of data and see the relationship directly without the need to move from one section to the other.

Comment 21: State the limitation of the study and possible protective mechanisms, aided by a schematic diagram.

Response 21: We find your suggestion very interesting. In fact, this aspect has already been explored in the last section of the discussion, in which the histological generalities of gastric ulcer development are mentioned; necrosis, edema, leukocyte infiltration, ischemia, angiogenesis, mucus secretion, bicarbonate, PG overproduction, and consequently inflammatory process (lines 268-275). In this context, a focus for future research would be to study the effect of PBPs on inflammatory processes derived from the development of gastric ulcer. Since the main limitation of our study is not having delved into the mechanisms of PBPs in the inflammatory process in gastric ulcers. To make our original idea more oriented towards your suggestion, we have added the information and the figure that you rightly suggest (Figure 5). So, we believe that now our text is clearer (lines 301-321).

Round 2

Reviewer 2 Report

 I accept in present form.